# Rapid Whole-Exome Sequencing as a Diagnostic Tool in a Neonatal/Pediatric Intensive Care Unit

**DOI:** 10.3390/jcm9072220

**Published:** 2020-07-13

**Authors:** Robert Śmigiel, Mateusz Biela, Krzysztof Szmyd, Michal Błoch, Elżbieta Szmida, Paweł Skiba, Anna Walczak, Piotr Gasperowicz, Joanna Kosińska, Małgorzata Rydzanicz, Piotr Stawiński, Anna Biernacka, Marzena Zielińska, Waldemar Gołębiowski, Agnieszka Jalowska, Grażyna Ohia, Bożena Głowska, Wojciech Walas, Barbara Królak-Olejnik, Paweł Krajewski, Jolanta Sykut-Cegielska, Maria M. Sąsiadek, Rafał Płoski

**Affiliations:** 1Department of Pediatrics, Division Propaedeutic of Pediatrics and Rare Disorders, Wroclaw Medical University, 50-368 Wroclaw, Poland; robert.smigiel@umed.wroc.pl (R.Ś.); michal.bloch88@gmail.com (M.B.); 2Lower Silesia Children’s Hospice, 51-163 Wroclaw, Poland; krzysztof.szmyd@formuladobra.pl; 3Department of Genetics, Wroclaw Medical University, 50-368 Wroclaw, Poland; e.szmida@gmail.com (E.S.); pawel.skiba@umed.wroc.pl (P.S.); maria.sasiadek@umed.wroc.pl (M.M.S.); 4Department of Medical Genetics, Warsaw Medical University, 02-106 Warsaw, Poland; walczak.m.anna@gmail.com (A.W.); piotr.gasperowicz@gmail.com (P.G.); joan.kosinska@wp.pl (J.K.); mrydzanicz@wum.edu.pl (M.R.); stawinski84@gmail.com (P.S.); biernackann@gmail.com (A.B.); rploski@wp.pl (R.P.); 5Postgraduate School of Molecular Medicine, Warsaw Medical University, 02-091 Warsaw, Poland; 6Department of Anesthesiology and Intensive Care, Wroclaw Medical University, 50-556 Wroclaw, Poland; marzena.zielinska@umed.wroc.pl (M.Z.); waldemar.golebiowski@umed.wroc.pl (W.G.); 7Department of Neonatology, Wroclaw Medical University, 50-556 Wroclaw, Poland; agnieszka.jalowska@gmail.com (A.J.); barbara.krolak-olejnik@umed.wroc.pl (B.K.-O.); 8Department of Neonatology, Provincial Specialist Hospital, 51-124 Wroclaw, Poland; grazynaohia@tlen.pl; 9Department of Anesthesiology and Intensive Care Unit for Newborns and Children, Provincial Specialist Hospital, 51-149 Wroclaw, Poland; glowska@poczta.onet.pl; 10Pediatric and Neonatal Intensive Care Unit, University Hospital, 45-401 Opole, Poland; wojciechwalas@wp.pl; 11Department of Forensic Medicine, Warsaw Medical University, 02-007 Warsaw, Poland; pkrajewski@wum.edu.pl; 12Department of Inborn Errors of Metabolism and Pediatrics, Institute of Mother and Child, 01-211 Warsaw, Poland; jolanta.cegielska@imid.med.pl

**Keywords:** WES, pediatric intensive care unit, genetic disorders, NGS, novel diseases, first-tier tests, mitochondrial disorders

## Abstract

Genetic disorders are the leading cause of infant morbidity and mortality. Due to the large number of genetic diseases, molecular and phenotype heterogeneity and often severe course, these diseases remain undiagnosed. In infants with a suspected acute monogenic disease, rapid whole-exome sequencing (R-WES) can be successfully performed. R-WES (singletons) was performed in 18 unrelated infants with a severe and/or progressing disease with the suspicion of genetic origin hospitalized in an Intensive Care Unit (ICU). Blood samples were also collected from the parents. The results from the R-WES were available after 5–14 days. A conclusive genetic diagnosis was obtained in 13 children, corresponding to an overall diagnostic yield of 72.2%. For nine patients, R-WES was used as a first-tier test. Eight patients were diagnosed with inborn errors of metabolism, mainly mitochondrial diseases. In two patients, the disease was possibly caused by variants in genes which so far have not been associated with human disease (*NARS1* and *DCAF5*). R-WES proved to be an effective diagnostic tool for critically ill infants in ICUs suspected of having a genetic disorder. It also should be considered as a first-tier test after precise clinical description. The quickly obtained diagnosis impacts patient’s medical management, and families can receive genetic counseling.

## 1. Introduction

Genetic conditions (single-gene disorders and copy number variants) are the leading cause of infant morbidity and mortality [1]. Many of these conditions are very rare (1:50,000 to 1:20,000 live births), but because of the high number of individual entities (approx. 8000 currently recognized syndromes according to OMIM https://www.omim.org/ or Orphanet https://www.orpha.net/consor/cgi-bin/index.php), the total number of affected individuals is large [2,3]. The impact of accurate and early genetic diagnosis is invaluable for the later clinical management of the patient and the family. Firstly, if any treatment is available, it could be implemented, allowing at the same time the avoidance of other ineffective or potentially harmful therapies. Secondly, long-term strategies (surgeries, rehabilitation etc.) could be scheduled to prevent complications of the disease. Thirdly, in cases with critically adverse prognosis, the diagnosis may be useful in discussing end-of-life decisions with the family. Finally, the knowledge of the molecular cause of the patient’s condition greatly facilitates further genetic counselling for the family [4].

Some of the genetic syndromes associated with severe symptoms in the neonatal period and infancy are readily diagnosed using targeted tests when symptoms are characteristic of a specific disorder. However, the targeted approach is difficult for many congenital conditions that are genetically heterogeneous or have no defined cause. Moreover, the main challenges are due to the critical illness and the early age of onset what means that some patients may not have fully grown into their phenotypes to make a clear clinical diagnosis. Moreover, many traditional diagnostic methods, even if effective, are too slow to provide useful information in severely ill patients at the Intensive Care Unit (ICU).

Over the past few years, Next Generation Sequencing (NGS), with its two main applications—whole exome sequencing (WES) and whole genome sequencing (WGS)—has started to be widely used for genetic analyses in ICUs for newborns and infants, especially when applied in the rapid mode. In critically ill infants, rapid-NGS can provide the results within 48–72 h and, when the clinical examination and interview are precisely collected, the diagnostic rates can reach >50% [5,6,7,8,9,10,11,12]. 

Here, we present our experience in the use of rapid-WES as a diagnostic tool applied as a first-choice examination in critically ill children at the ICU.

## 2. Materials and Methods

### 2.1. Patients Recruitment

All the parents signed a written informed consent form for the genotyping and consented to the publishing of all the data generated. The study received the approval of the Bioethics Committee of Wroclaw Medical University (code: KB-430/2018; date of approval: 23 July 2018) and was conducted in research settings.

Data were analyzed from children—patients of ICUs in the city of Wroclaw (Poland) who were consulted by geneticists in the years 2015–2019. A decision to perform rapid-WES (R-WES) was made for 18 unrelated infants during their ICU stay (10 patients from newborn ICUs and 8 from pediatric ICUs) with a severe and/or progressive disease, a suspicion of genetic origin, and who have met the inclusion criteria presented in Table 1.

After performing R-WES, out of these children, 13 (72.2%) obtained a molecular diagnosis explaining their clinical condition. R-WES was the first genetic testing for nine patients. For the other nine patients, R-WES was performed after previous (pre- and postnatal) genetic tests, such as karyotyping, array comparative genomic hybridization (CGH), or target molecular tests for a single gene or a panel of genes using classical Sanger sequencing or the NGS technique.

### 2.2. Genetic Analysis 

Venous blood samples were collected from 18 children and from their parents and siblings (if present). Genomic DNA was extracted using a standard protocol.

R-WES was defined as a process completed within 5–14 days of the sample collection and included transport to the laboratory, DNA isolation, sequencing, and the first analysis of the WES results. WES was performed on the proband DNA using SureSelect Human All Exon v5 (16 patients) or v7 (two patients) (Agilent Technologies, Palo Alto, CA, USA) according to the manufacturer’s instructions. The libraries were paired-end sequenced (2 × 100 bp) on the HiSeq 1500 (Illumina, San Diego, CA, USA) in Rapid Run mode and analyzed as previously described [13].

The variants considered as disease causing were validated in proband and studied in all the available family members by direct Sanger sequencing using the BigDye Terminator v3.1 Kit (Applied Biosystems, Foster City, CA, USA) on the ABI 3500Xl Genetic Analyzer (Applied Biosystems), or by amplicon deep sequencing (ADS) performed using the Nextera XT Kit (Illumina) and sequenced on the HiSeq 1500 (Illumina).

### 2.3. WES Data Analysis

The quality control of the raw fastq reads was performed, followed by adapter trimming and the removal of low quality reads using Trimmomatic [14]. Burrows-Wheeler Alignment tool (BWA) was used to map the reads on hg38, followed by sorting and duplication removal using samblaster [15,16]. Variant identification was performed using multiple algorithms: HaplotypeCaller from GATK, Freebayes, DeepVariant, and MuTect2 [17,18,19,20]. Structural variants were identified using Lumpy [21]. Copy number variation (CNV) identification was performed using CNVKit [22]. The identified variants were annotated using the Ensembl VEP as well as multiple databases, including dbSNP, dbNSFP, GnomAD, ClinVar, and HGMD [23,24,25,26,27,28]. Moreover, an inhouse database of Polish WES (*N* > 2000) was used to identify the sequencing artifacts as well as the variants common in the Polish population.

All the variants were filtered to include those with a frequency of <0.01 and a predicted effect on the protein sequence (unless they were already annotated as pathogenic in ClinVar or HGMD). The filtered variants were manually inspected and evaluated against the patient’s phenotype and the ACMG pathogenicity criteria, as implemented in Varsome [29,30]. There were no incidental findings eligible for reporting [31]. A supplementary table with the variants pathogenicity criteria is available in the supplementary files (Appendix A).

## 3. Results

### 3.1. Clinical Characteristics of Patients 

The clinical characteristics of the children who went through the R-WES are summarized in Table 2. Among the patients, there were 11 males and 7 females, aged between 3 days and 16 months at the moment when R-WES was performed. Eight patients were newborns less than one month old; four patients were in the age group 2–6 months old; in the next group of 7–12 months old, there were five patients and only one patient older than one year.

Fourteen out 18 patients (77.7%) died—for seven patients, the WES results came postmortem.

In eight cases, the pregnancy was affected by polyhydramnios (3/8), weak fetal movements (2/8), fetal hydrops (2/8), and intracranial cysts (1/8). The prenatal period in the other presented cases was normal.

### 3.2. Genetic Findings in WES

The R-WES was performed among 18 patients suspected of having a genetic disease who were in a severe condition. The preliminary WES results were available after 5–14 days.

A conclusive molecular diagnosis was made in 13 out of 18 proband children identified by WES, corresponding to an overall diagnostic yield of 72.2%. All the variants were confirmed by direct Sanger sequencing or ADS. The identified genes and sequence variants are presented in Table 2. Eight patients were diagnosed with inborn errors of metabolism (IEM), especially mitochondrial diseases (seven patients) and among them were three patients with *SCO2* gene variants. The following disorders from the IEM group were diagnosed: Leigh syndrome (OMIM:604377) in three patients, Alpers syndrome (OMIM:203700), pyruvate carboxylase deficiency (OMIM:216150), Anderson syndrome (OMIM:232500), and combined oxidative phosphorylation deficiency 6 (OMIM:616974), as well as one patient with a suspicion of a mitochondrial disease who had two heterozygous variants in *TRMT10C* (combined oxidative phosphorylation deficiency 30, OMIM:616974). In addition, the following diseases (five cases) have been diagnosed: Schaaf–Yang syndrome (OMIM:615547), hypotonia infantile with psychomotor retardation and characteristic facies 1 syndrome (IHPRF1, OMIM:611549), surfactant metabolism dysfunction pulmonary 3 (SMPD3, OMIM:610921), nemaline myopathy (OMIM:161800), and neurofascin defect (OMIM:618356).

In two patients, we found possibly causative variants in genes with a known function, but without previous associations with human disease: *DCAF5* and *NARS1*.

#### 3.2.1. *DCAF5* Variant

Considering the phenotype and the characteristics of all the variants, as a disease-causing mutation, a heterozygous stop-gain variant in *DCAF5* (hg38: g.14:069055385-G>C; NM_003861.3:c.1301C>G; p.(Ser434Ter) was prioritized. The identified variant has 0 frequency in all tested databases (gnomAD, EXaC, ESP6500, 1000 genomes, as well as an in-house database of >2000 Polish exomes). The variant may cause a loss of function, as it is predicted to significantly truncate the protein (it occurs at aminoacid position 434 out of 943). The *DCAF5* gene has a high pLI score (1), with an o/e (loss-of-function observed/expected) score = 0 and an LOEUF (loss-of-function observed/expected upper bound fraction) score = 0.08, which indicates intolerance to monoallelic loss of function [26]. Intriguingly, in the gnomAD database, which includes >140,000 samples, there are no loss-of-function variants reported for *DCAF5*. However, as the premature stop codon is located in the 9th, i.e., the last exon of the gene, the truncated DCAF5 protein may yet be expressed, perhaps exerting a dominant negative effect. A family study performed by ADS confirmed the presence of the c.1301C>G variant in the patient sample and excluded its presence in the patient’s parents, which suggests a de novo mutation. (Figure 1a). The parenthood in the family was confirmed using a panel of dedicated forensic short tandem repeat (STR) markers.

#### 3.2.2. *NARS1* Variant

In the second patient, we prioritized the homozygous missensse mutation in the *NARS1* gene (hg38: g.chr18-57606713-A>G; NM_004539.3:c.1040T>C; p.Phe347Ser). This variant was absent in the databases mentioned above and has high and consistent pathogenicity prediction scores (nine pathogenic predictions from DEOGEN2, EIGEN, FATHMM-MKL, M-CAP, MVP, MutationAssessor, MutationTaster, REVEL, and SIFT vs. one benign prediction from PrimateAI) [32]. The *NARS1* gene is likely to be associated with a recessive disease, as indicated by the pRec = 0.9969 (probability of being intolerant to homozygous loss of function variants) [33]. A family study performed by ADS confirmed the presence of the c.1040T>C variant in the patient sample and revealed the carrier status for both the healthy parents. ADS was also applied to the patient’s healthy siblings (brother and sister) and excluded the presence of the variant in both of them (both shown to be wild-type homozygous, Figure 1b).

#### 3.2.3. *NFASC* Variant

The homozygous variant NM_015090.3:p.(Arg831*)/c.2491C>T in *NFASC* was described in detail in our previous paper [34].

In three patients, the molecular diagnosis remains unknown. For those patients, we are planning to reanalyze the WES data or extend the genetic tests (arrayCGH, WGS).

## 4. Discussion

A significant number of genetic diseases start are revealed in the first year of life. For the patient and his family, the “diagnostic odyssey” usually begins with the first symptoms of the suspected genetic disease. A diagnosis with serial gene sequencing or with non-genetic testing may be time-consuming and aggravating for the patient, and it frequently brings no result in terms of finding the reason for the disease during the ICU stay.

Our clinical sample is limited in size, but what enhances our work is the diagnostic utility and high mortality rate [35,36,37]. The diagnostic yield in our results (72.2%) is higher than that previously reported by studies using NGS [9,12,35,36,37,38,39,40]. The difference in diagnostic yield may result from a statistical fluctuation due to the relatively low number of patients. However, the main reason may be the severity of clinical condition in our cohort, exemplified, among others, by the high death rate of 77.7%. Notably, while in the Meng et al. study, the patients had more diverse disease in terms of severity, in the group of most seriously ill infants, the diagnostic yield was significantly higher than in the other groups [40].

We have also shown that R-WES is successful as a first-tier test for patients suspected of having a genetic disorder with a progressing course—88.9% (8/9) of examined infants for whom R-WES was the first-tier test received a molecular diagnosis.

In this study, the cohort of IEM was found in eight patients and, consistent with the literature, it was mainly caused by mitochondrial disorders (7/8) [41,42].

Many clinical conditions such as IEM or neurodegenerative diseases are still unexplained, and the molecular background remains unknown. This could be, in particular, the case for the most severe conditions, in which a lethal ending makes the diagnosis difficult. In keeping with this, in our study we have found possibly causative variants in three genes so far not associated with human diseases: *NARS1, DCAF5,* and *NFASC*.

A homozygous *NFASC* variant rs755160624 predicted to cause the selective loss of the glial-specific Nfasc155 isoform was found in a child with a severe and progressive neurological disease. The variant, clinical phenotype, and related immunofluorescence study confirming the absence of the Nfasc155 isoform in the patient have been published separately as a description of a novel Mendelian disease [34]. Recently, our report has been confirmed by a description of a series of patients with the *NFASC* disorder [43].

Another candidate for novel gene-disease association is the *NARS1* gene, in which we have found a homozygous ultrarare missense mutation that is likely pathogenic. *NARS1* encodes arginine aminoacyl-tRNA synthetase, which is critical for the process of translation. The NARS protein belongs to a family of aminoacyl-tRNA synthetases whose many members (but not NARS) have been associated with severe and progressive diseases affecting the nervous system and inherited in an autosomal recessive mode [44,45]. Given the general consistency of our proband’s symptoms with those caused by defects of other aminoacyl-tRNA synthetases, it is possible that our patient is the first case of human disease caused by the *NARS1* variant. The proband also had a sibling who died earlier with similar symptoms, but he could not have been tested due to the lack of good quality DNA obtained from the paraffin tissue blocks stored during the postmortem examination. Thus, although we found an interesting candidate gene, further research is needed to confirm the clinical meaning of the *NARS1* defect.

The least-known gene whose variant we propose to be causative of the disease in one of our probands is the *DCAF5* gene, in which we found a de novo variant causing a premature stop codon. In the literature, there are only three patients described with deletions in 14q24.1q24.3, which includes also the *DCAF5* gene [46]. Patients with the mentioned deletion presented mild intellectual disability, congenital heart defects, brachydactyly, and additional abnormalities, but none of them had a severe and progressive condition [46]. Our patient had a serious, severe, and lethal disorder with the following symptoms: respiratory insufficiency; no muscle tension; and weak reflexes, including the sucking reflex; residual spontaneous motoric (weak movement of upper and lower limbs); no eye contact; symmetric extensor dominance; areflexia; the absence of motor and oral automatism; as well as chest deformation. Since the stop codon created by the mutation in our patient occurred in the last exon of the gene, we speculate that the severe phenotype may be caused by both the *DCAF5* haploinsufficiency (due to the truncation of >50% of the protein) and a dominant negative effect possibly exerted by the truncated protein (which is likely expressed, given that the stop codon is located in the last exon and thus should not undergo NMD (Nonsense-Mediated Decay)). DDB1-and-CUL4 associated factor 5 is the product of the brain-expressed *DCAF5* gene, which functions as a component of the ubiquitin ligase complex *DDB1-CUL4* [47,48]. A major part of this complex is *CUL4B*, whose mutations cause X-linked intellectual disability [49]. We propose that the dysfunction of its binding partner *DCAF5* caused by the de novo mutation in our patient might have similar consequences for developmental function. However, clearly further research is needed to explore the pathogenicity of the *DCAF5* defect.

In this study, we have not analyzed the economic issue, but it is highly possible that, thanks to R-WES used as a first- or second-tier test in our patients, we have avoided time and money-consuming procedures and also significantly shortened the time of hospitalization for patients who could be discharged to be kept under the care of a home-hospice (4/18, 22.2%).

### Impact of WES on Clinical Management

For patients with severe conditions, a molecular characterization may have fundamental practical implications. Finding a disease-causing mutation responsible for patients’ clinical condition is an important aspect for the patient’s family. In cases when specific treatment is available, we can provide it and ward off in the same time ineffective, empirical, or detrimental therapies. Moreover, knowledge about the etiology and the natural history of the disease could also be useful in cases with an unfavorable prognosis in order to discuss with the family the most appropriate end-of-life decisions. Finally, after examining parents, a full genetic consultation could be offered to the family and help the parents with further reproductive choices.

As mentioned earlier, in 13 out of 18 of our patients a molecular diagnosis was achieved, and in 2 patients there were found variants which could be a possible cause of their symptoms. For all of their families, a genetic consultation was prepared—in 80% of cases (12/15), the carrier state was found in both healthy parents or a healthy mother; for the others (3/15), the pathogenic variants were de novo. For five patients, after receiving the WES results, “Do Not Resuscitate” protocols were signed based on the clinical conditions of the patients; another three were covered by hospice palliative care. Three patients had multidisciplinary care offered, including lung transplantation in patient number 7. Seven patients died before the WES results were available.

In nine cases, R-WES was the first-choice tool, with a diagnostic yield of 88.9%. In other cases, WES was preceded by karyotyping (two prenatal and one postnatal), aCGH (one prenatal and three postnatal), and target molecular tests for chosen mutations in a single gene or in a panel of genes (four cases).

## 5. Conclusions

Rapid-WES is an effective and time-saving diagnostic tool for infants and children in ICUs who present heterogeneous and severe symptoms. In this specific group of patients, where there is no time to lose and the diagnostic options are often limited because of the patient’s age and severe condition, rapid-WES should be considered even as a first-tier test.

## Figures and Tables

**Figure 1 jcm-09-02220-f001:**
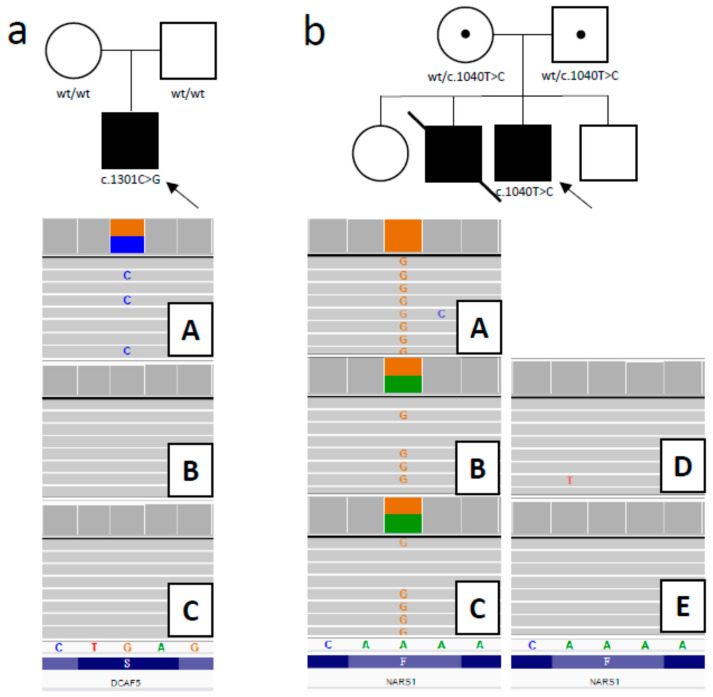
Family study results. (**a**) Heterozygous stop-gained mutation c.1301C>G in *DCAF5* was confirmed in patient (**A**) and excluded in the patient’s parents (**B**—mother, **C**—father) (family on the left). (**b**) Homozygous missensse mutation c.1040T>C in *NARS1* (**A**— patient). Parents proved to be carriers (**B**—mother, **C**—father), and the siblings are wild-type homozygotes (**D**—brother, **E**—sister) (family on the right). Circle represents female, square indicates male, filled symbol indicates affected individual, proband is marked with a black arrow; wt—wild-type.

**Table 1 jcm-09-02220-t001:** Inclusion and exclusion criteria in the presented study.

Inclusion Criteria (All the Following)	Exclusion Criteria (Any of the Following)
A critically ill newborn or infant in the ICU with severe unexplained neurological signs that started suddenly, but the following conditions also will be considered: metabolic failure of unknown origin; severe multi-organ disease of unknown pathogenesis, especially in case of poor responsiveness to standard treatment; severe congenital malformations that are not consistent with any known syndrome; other unexplained or unclear acute conditions.	Presence of symptoms suggesting a concrete, known genetic syndrome possible to diagnose using standard diagnostic methods (for example, Smith–Lemli–Opitz syndrome or congenital metabolic defect diagnosed in a newborn screening test performed in Poland (MS/MS, 25 inborn errors of metabolism).
	Based on pre- and perinatal history, a non-genetic etiology can explain the disease, and/or is confirmed with laboratory results/imaging techniques.
Consent form obtained from both parents for blood sampling and genetic research tests of the child and themselves.	Lack of consent form of one of the proband’s parents for blood sampling and genetic research test.

**Table 2 jcm-09-02220-t002:** Clinical characteristics, identified genes, and sequence variants in the patients. M—male; F—female; RF—respiratory failure; D—dysmorphia; FH—flaccidity/hypotonia; MA—metabolic acidosis; Y—yes; N—no; AR—autosomal recessive mode of inheritance; AD—autosomal dominant mode of inheritance; XLR—X-link recessive mode of inheritance.; DNR protocol—Do Not Resuscitate protocol.

ID/Sex	Age of Onset/Age at Testing	Main Symptoms	Rapid-WES as First-Tier/Molecular Confirmation in WES	Gene/Inheritance	Variant(s) hg38	Pathogenicity Verdict according to ACMG Classification (https://varsome.com)	Disease/ OMIM	Impact on Clinical Management	Death
1/M	3^rd^/ 7^th^ week of life	RF; distal contractures;	N/Y	*SCO2* AR	compound heterozygote NM_005138.3:		Leigh syndrome/604377	Death before diagnosis	Y
					chr22:050524395-C > CTGAGTCACTGCTGCATGCT c.16_17insAGCATGCAGCAGTGACTCA; p.(Arg6GlnfsTer82)	Pathogenic			
					chr22: 50523994-C > T c.418G > A; p.(Glu140Lys)	Pathogenic			
2/M	5^th^/ 8^th^ month of life	RF; FH; D; MA;	Y/Y	*SCO2* AR	homozygote NM_005138.3: chr22: 50523994-C > T c.418G > A; p.(Glu140Lys)	Pathogenic	Leigh syndrome/604377	Palliative hospice care	Y
3/M	3^rd^/ 10^th^ month of life	RF; FH;	N/Y	*POLG* AR	homozygote NM_001126131.2: chr15:89320885-G > C c.2862C > G; p.(Ile954Met)	Likely Pathogenic	Alpers syndrome/203700	Palliative hospice care	Y
4/F	Prenatal period/ 3^rd^ week of life	RF, arthrogryposis; D;	N/Y	*GBE1* AR	compound heterozygote NM_005158.3:		Glycogen storage disease IV - perinatal severe form (Anderson syndrome)/232500	Death before diagnosis	Y
					Chr3:81648854-A > G c.691+2T > C; p.-	Pathogenic			
					chr3:81499187-C > T c.1975G > A; p.(Gly659Arg)	Uncertain Significance			
5/F	1^st^ day of life/ 2^nd^ month of life	RF; MA; D;	N/Y	*PC* AR	compound heterozygote NM_022172.3:		Pyruvate carboxylase deficiency/ 216150	Death before diagnosis	Y
					chr11:66866282-G > A c.1090C > T; p.(Gln364Ter)	Pathogenic			
					chr11:66863920-C > G c.1222G > C; p.(Asp408His)	Pathogenic			
6/M	2^nd^/ 3^rd^ month of life	RF; D; epilepsy; brain atrophy	Y/Y	*AIFM1* XLR	hemizygote NM_004208.3: chrX:130133411-C > G c.1350G > C; p.(Arg450Ser)	Pathogenic	Combined oxidative phosphorylation deficiency 6/ 300816	Palliative care, DNR protocol	Y
7/F	4^th^/ 16^th^ month of life	RF;	Y/Y	*ABCA3* AR	homozygote NM_001089.3: chr16:002323532-C > T c.604G > A; p.(Gly202Arg)	Uncertain Significance	Surfactant metabolism dysfunction, pulmonary, 3 (SMPD3)/ 610921	Lung transplantation	N
8/F	At birth/ 2^nd^ month of life	RF, arthrogryposis; D; FH;	N/Y	*MAGEL2* AD	de novo NM_019066.4: chr15:23644849-C > T c.2894G > A; p.(Trp965Ter)	Pathogenic	Schaaf-Yang syndrome/ 615547	symptomatic treatment, multidisciplinary care	N
9/F	1^st^/ 8^th^ month of life	FH; severe delayed psychomotor development; D;	Y/Y	*NALCN* AR	compound heterozygote NM_052867.2:		Hypotonia, infantile, with psychomotor retardation and characteristic facies 1 (IHPRF1)/ 611549	symptomatic treatment, multidisciplinary care	N
					chr13:101111216-G > A c.2203C > T; p.(Arg735Ter)	Pathogenic			
					Chr13:101229388-C > A c.1626+5G > T; p-	Uncertain Significance (note: predicted to strongly affect splicing -ADA Score 0.9997. If this is taken into account the verdict is „pathogenic”).			
10/F	At birth/ 1^st^ month of life	RF; scleroderma;	N/Y	*ACTA1* AD	de novo NM_001100.3: Chr1:229432567-C > A c.443G > T, p.(Gly148Val)	Likely Pathogenic	Nemaline myopathy AD/ 161800	Death before diagnosis	Y
11/M	Prenatal period/ 1^st^ week of life	scleroderma edema;	N/N	non-diagnostic	-		lack	Death before diagnosis	Y
12/M	1^st^/ 9^th^ day of live	RF; MA; elevated alanine	Y/Y	*TRMT10C* AR	compound heterozygote NM_017819.4:		Mitochondrial disease (Combined oxidative phosphorylation deficiency 30)/ 616974	Palliative care, DNR protocol	Y
					Chr3:101565509-T > C c.728T > C; p.(Ile243Thr)	Uncertain Significance			
					Chr3:101565164-C > CA c.393_3394insA; p.(Tyr132IlefsTer15)	Likely pathogenic			
13/F	At birth/ 3^rd^ day of life	RF; F;	N/Y	*NFASC* AR	homozygote NM_015090.3: Chr1:204984059-C > T c.2491C > T; p.(Arg831Ter)	Pathogenic	New disorder described (Neurodevelopmental disorder with central and peripheral motor dysfunction)/ 618356	Palliative care, DNR protocol	Y
14/M	11^th^/ 11^th^ month of life	RF; encephalopathy; sister died earlier with similar signs;	Y/Y	*NARS1* AR	homozygote NM_004539.4: Chr18:57606713 -A > G c.1040T > C; p.(Phe347Ser)	Uncertain Significance	A new disease suspected/108410	Palliative care, DNR protocol	N
15/M	1^st^ week of life/ 4^th^ month of life	RF; F; D;	Y/Y	*DCAF5* AD	de novo NM_003861.3: 14:069055385-G > C c.1301C > G; p.(Ser434Ter)	Pathogenic	A new disease suspected/603812	Palliative care, DNR protocol	Y
16/M	2^nd^/ 7^th^ day of life	Hyperammonemia; RF;	Y/N	non-diagnostic	-		lack	Death before diagnosis	Y
17/M	Prenatal period/ 1^st^ month of life	General eodema; seizures;	N/N	non-diagnostic	-		lack	Death before diagnosis	Y
18/M	2^nd^/ 8^th^ month of life	Psychomotor delay, severe deterioration of general condition	Y/Y	*SCO2* AR	homozygote NM_005138.3: chr22: 50523994-C > T c.418G > A; p.(Glu140Lys)	Pathogenic	Leigh syndrome/604377	Palliative hospice care	Y

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
