# Peer review of "Rapid Whole-Exome Sequencing as a Diagnostic Tool in a Neonatal/Pediatric Intensive Care Unit"

_jcm, 2020, doi:10.3390/jcm9072220_

Round 1

Reviewer 1 Report

The authors should be commended on this interesting and well-written study.

However, one major critique is around patient selection – this is particularly a relevant point as the study found that 83.3% of cases had a molecular diagnosis, which is much higher than most studies. The authors should clarify their inclusion and exclusion criteria. Otherwise, it is not possible to comment on if there was any ascertainment bias in their case selection.

Could you please clarify the timing of the whole exome sequencing (WES)? Were the patients all sequenced during their ICU stay? Was this the first presentation of their disease?

Rapid-WES is stated to have the data analyzed within 72 hours. Why were results not available until 5-14 days. To clarify the 66.7% deaths all occurred in less than 14 days from the time rapid-WES was sent?

Of minor note, there are some more colloquial expressions that may benefit from editing from a native English speaker (e.g. perhaps "long-term" instead of "long-distance" line 32 and “gestalt diagnosis, dysmorphology pearls” line 38).

Reviewer 2 Report

Reconsider after Major Revisions

In the manuscript entitled, “Rapid whole-exome sequencing as a diagnostic tool in a neonatal/pediatric intensive care unit”, the authors describe the results of rapid whole exome sequencing in 18 patients. It is a descriptive report that while not necessarily novel, potentially serves a purpose by adding to the growing body of literature supporting the use of this technology for diagnosis of monogenic disease in a critically ill pediatric population. However, the authors make several conclusions that are not fully supported and the manuscript is hampered by numerous grammatical errors that sometimes make it challenging to understand. The most significant criticism is that novel variants in two genes are described as causal new disease-causing genes without sufficient detail as to why they are considered likely pathogenic. One strength of this manuscript is the high mortality rate among this cohort of patients. Numerous patients went on to DNR or hospice care after diagnosis. However a major criticism is that there is no data regarding how the genetic diagnosis affected the clinical management. Seven patients died prior to WGS resulting – perhaps all these patients would have been DNR or hospice regardless of genetic diagnosis?

Major comments:

  1. This manuscript has numerous grammatical and other English language errors and would greatly benefit from careful editing and review.
  2. One of the major issues with the authors’ conclusions is their decision to count NARS and DCAF5 in their diagnostic yield, as these have not been associated with human disease and the authors certainly have not made a compelling argument in this manuscript for a new gene-disease association.
    1. Specifically, “conclusive genetic diagnosis in 15 children” (line 12) is inaccurate. This number should be revised to 13 and the overall yield to 13/18 or 72%.
    2. DCAF5: The authors describe a de novo variant but there is insufficient evidence to conclude that this variant explains the patient’s phenotype. Reference 20 lists 18 deleted genes in addition to DCAF5 and as none of them had a severe progressive condition, this actually argues against DCAF5 being disease-causing in your patient. The rationale that is proposed for DCAF5 as causative of this patient’s phenotype is tangential and unfounded.
    3. NARS: (lines 191-192) it is an unfounded conclusion to state the patient is the first case of human disease caused by AR missense mutations in NARS. Additional information is required to conclude that this is a causal variant. It would be acceptable to say “we found an interesting candidate gene that we plan on exploring”
  3. The criteria for study inclusion are unclear. What criteria did the geneticists use to select the patients for WES? (line 56) I imagine that over a 4 year period there were more than 18 patients who were critically ill in the ICU- how was the decision made to choose these 18?
  4. Did these patients receive standard newborn screening? If so, did that identify any of these disorders? Many readers will not have familiarity with what is on the Polish newborn screen.
  5. An additional table is needed with the variant scoring criteria that were used. It is unclear to the reader that a variant would score out as likely pathogenic without any previous association with human disease or any gene-disease relationship.
  6. The authors should strongly consider analyzing clinical utility rate as well- most recent publications on WGS/WES in the ICU are examining changes in clinical management in addition to only diagnostic yield.

Specific/minor comments:

  1. How many patients were in the NICU versus the PICU?
  2. How were the variants prioritized in the analysis platform?
  3. I do not understand what you mean in lines 38-39 by gestalt diagnosis and dysmorphology pearl s- how are these examples of targeted tests?
  4. Line 47: references 5 and 6 for diagnostic yield are both for WGS, not WES. There are many WES studies that could be cited.
  5. “mostly patients of ICUs in Wroclaw”- are the other patients not in the ICU? Or they are in an ICU in a different city?
  6. Were there any incidental findings? How were these addressed?
  7. Line 67: you are saying you define RWES as preparation and analysis in 48-72 hours but in your abstract you say the results took 5-14 days. Is this discrepancy because of Sanger confirmation? Please clarify the time course.
  8. Line 84: says that ten patients had severe and rapidly progressive conditions and 8 patients had progressing and severe conditions but the course of the disease was moderate. But then 14 out of 18 died. How is the course of the disease moderate if some of these 8 patients died?
  9. Line 87: 14 out of 18 is 77.7% not 66.7%
  10. Line 89: were the other pregnancies normal? This may be more appropriate as an additional column in your table instead of here
  11. Line 99: what are “first” WES results? Do you mean preliminary?
  12. Line 100: It would be more appropriate to say that a molecular diagnosis was made
  13. Line 110: says the following five cases have been diagnosed but only lists four diseases
  14. Line 154: what does extend the genetic tests mean? Do you mean WGS?
  15. Line 156: are you trying to say that WES should only be done on children less than 1? What is the purpose of this sentence?
  16. Line 161: I disagree that a distinguishing feature of this manuscript is that the patients have severe disease requiring ICU care (PMIDs 30049826, 29644095, 31246743)- mortality rate is acceptable as a distinguishing feature
  17. Line 166: if mortality is 14/18 that is 77.7%
  18. Line 164: how would a low number of patients change the diagnostic yield?
  19. Line 166 “notably, while in other studies patients had been more diverse…” what do you mean by diverse? In their ethnic background? Disease presentation? Then you only cite one study, not “studies”
  20. Line 168: further increase your diagnostic yield compared to what? The Meng study that you cite in the previous line was also WES this is confusing
  21. Line 173: how do you say mainly caused by mitochondrial disease when it is 46.7%? This sentence doesn’t seem to add to the purpose of the paper
  22. Line 210: you have not demonstrated that your patient lacked cognitive function- they were an infant
  23. Line 238: are you limiting this just to infants?
  24. Table: what are the asterisks indicating in patients 4, 5, 8, 9, and 13
  25. Consider changing “lack” in column disease/omim for the undiagnosed patients to something like “non-diagnostic”
  26. These sentences in particular don’t make sense in English and need to be rewritten
    1. Line 17: “presented heterogeneous and severe symptoms”
    2. Line 47: “clinical examination and interview are precisely collected”- are you trying to say that your analysis pipeline is phenotype-driven?
    3. Line 176: fast-fatal ending
    4. Line 200: vestigial spontaneous motoric
    5. Line 218: provide a diagnostic response to their families
    6. Lines 224-225
  27. These sentences are also poorly written
    1. Line 6: …genetic and phenotype heterogeneity and very often rapid course…
    2. Line 20: “allows to offer”
    3. Line 32: do you mean long-term?
    4. Line 41: do you mean severely ill?
    5. Line 65: question use of the word obligatorily
    6. Line 222: a full genetic consulting could be offered- do you mean consultation?
  28. There are a number of minor grammatical and punctuation errors throughout the manuscript

Author Response

Dear Reviewer,

Thank you for all your valuable comments and very careful analysis of the article. We have responded to every comment and we hope we have met your expectations.

Major comments:

Point 1: This manuscript has numerous grammatical and other English language errors and would greatly benefit from careful editing and review.

Response 1: The mentioned phrases were changed. The text was again corrected by native English speaker.

Point 2: One of the major issues with the authors’ conclusions is their decision to count NARS and DCAF5 in their diagnostic yield, as these have not been associated with human disease and the authors certainly have not made a compelling argument in this manuscript for a new gene-disease association.

Response 2: According to Reviewer’s suggestion, we have corrected the diagnostic yield to 72.2%  in all parts of the manuscript.

Point 2.1: Specifically, “conclusive genetic diagnosis in 15 children” (line 12) is inaccurate. This number should be revised to 13 and the overall yield to 13/18 or 72%.

Response 2.1: As above, we have corrected the diagnostic yield in all parts of the manuscript.

Point 2.2: DCAF5: The authors describe a de novo variant but there is insufficient evidence to conclude that this variant explains the patient’s phenotype. Reference 20 lists 18 deleted genes in addition to DCAF5 and as none of them had a severe progressive condition, this actually argues against DCAF5 being disease-causing in your patient. The rationale that is proposed for DCAF5 as causative of this patient’s phenotype is tangential and unfounded.

Response 2.2: We agree that the evidence for causal role of DCAF5 variant is weak and we have removed this patient from the list of patients with a molecular diagnosis (as well as the patient with the  NARS variant). However, we would like to argue that the full deletion may not be equivalent to our variant. As we write in the Discussion: “we speculate that the severe phenotype may be caused by both the DCAF5 haploinsufficiency (due to truncation of >50% of the protein) and a dominant negative effect possibly exerted by the truncated protein (which is likely expressed given that the stop codon is located in the last exon and thus should not undergo NMD)”.

However, to emphasize that these are only hypotheses we added  the sentence in the Discussion section: “Further research is needed to explore pathogenicity of DCAF5 defect.”

Point 2.3: NARS: (lines 191-192) it is an unfounded conclusion to state the patient is the first case of human disease caused by AR missense mutations in NARS. Additional information is required to conclude that this is a causal variant. It would be acceptable to say “we found an interesting candidate gene that we plan on exploring”

Response 2.3: Thank you for this comment. We added a sentence in the Discussion section: “The proband also had a sibling who died earlier with similar symptoms but he could not have been tested due to lack of good quality DNA obtained from paraffin tissue blocks stored during postmortem examination. Thus, although we found an interesting candidate gene, further research is needed to confirm the clinical meaning of NARS1 defect.” (Lines 226-230)

Point 3: The criteria for study inclusion are unclear. What criteria did the geneticists use to select the patients for WES? (line 56) I imagine that over a 4 year period there were more than 18 patients who were critically ill in the ICU- how was the decision made to choose these 18?

Response 3: Indeed, we have not specified the inclusion and exclusion criteria in the manuscript. We added them as a table (Table 1) in the section Materials and Methods:

Table 1. Inclusion and exclusion criteria in the presented study.

Inclusion criteria (all the following)

Exclusion criteria (any of the following)

a critically ill newborn or infant in the ICU with severe unexplained neurological signs  that started suddenly, but the following conditions also will be considered: metabolic failure of unknown origin; severe multi-organ disease of unknown pathogenesis, especially in case of poor responsiveness to standard treatment; severe congenital malformations that are not consistent with any known syndrome; other unexplained or unclear acute conditions;

presence of symptoms suggesting a concrete, known genetic syndromes possible to diagnose using standard diagnostic methods (for example Smith-Lemli-Opitz syndrome or congenital metabolic defect diagnosed in newborn screening test performed in Poland (MS/MS, 25 inborn errors of metabolism);

based on pre- and perinatal history, a non-genetic etiology can explain the disease, and/or is confirmed with laboratory results / imaging techniques

consent form obtained from both parents for blood sampling and genetic research tests of the child and themselves

lack of consent form of one of the proband’s parents for blood sampling and genetic research test

Point 4: Did these patients receive standard newborn screening? If so, did that identify any of these disorders? Many readers will not have familiarity with what is on the Polish newborn screen.

Response 4: All patients received standard newborn screening;  a positive result was an exclusion criterion (Table 1 “congenital metabolic defect diagnosed in newborn screening test performed in Poland (MS/MS, 25 inborn errors of metabolism)”.

Point 5: An additional table is needed with the variant scoring criteria that were used. It is unclear to the reader that a variant would score out as likely pathogenic without any previous association with human disease or any gene-disease relationship.

        Response 5: We added an additional column with Pathogenicity prediction according to ACMG classification              [Richards S., et al., 2015] (source https://varsome.com) to each variant in Table 2 (previously Table 1).

As recommended by the Reviewer, we have created an additional table with the variant scoring criteria. The table is large in size and also has references, so we suggest to add it as Supplementary Materials. We hope that the table will satisfy your comment.

Point 6: The authors should strongly consider analyzing clinical utility rate as well- most recent publications on WGS/WES in the ICU are examining changes in clinical management in addition to only diagnostic yield.

Response 6: The Reviewer has an excellent suggestion, but the analyzing the clinical utility rate is beyond the main scope of our article. Short information about clinical managements and outcomes are in the table 2. Additionally, in lines 263-269 we mentioned the management and important info about genetic family consultation: 

“As mentioned earlier, in 13 out of 18 our patients a molecular diagnosis was achieved and in two patients there were variants which could be a possible cause of symptoms. For families of all probands a genetic consultation was prepared – in 80% cases (12/15) the carrier state was found in both healthy parents or a healthy mother, for others (3/15) the pathogenic variants were de novo. For five patients, after receiving WES results, “Do Not Resuscitate” protocols were signed based on the clinical conditions of the patients, another three were covered by hospice palliative care. Three patients had a multidisciplinary care offered including lung transplantation in patient number 7. Seven patients died before the WES results were available.”

Specific/minor comments:

Point 1: How many patients were in the NICU versus the PICU?

Response 1: We have added this information in line 60, section 2.1. Patients recruitment:

“Data were analyzed from children, patients of ICUs in the city of Wroclaw (Poland) who were consulted by geneticists in the years 2015-2019. A decision of performing Rapid-WES (R-WES) was made for 18 unrelated infants during their ICU stay (10 patients from newborn ICUs and 8 from pediatric ICUs) with a severe and/or progressive disease, suspicion of genetic origin and which have met the inclusion criteria presented in Table 1..”

Point 2: How were the variants prioritized in the analysis platform?

Response 2: An extra section was added: “2.3. WES data analysis:

               The quality control of raw fastq reads was performed, followed by adapter trimming and low quality reads removal using Trimmomatic [14]. BWA was used to map reads on hg38, followed by sorting and duplication removal using samblaster [15,16]. Variant identification was done using multiple algorithms: HaplotypeCaller from GATK, Freebayes, DeepVariant and MuTect2 [17–20]. Structural variants were identified using Lumpy [21]. CNV identification was performed using CNVKit [22]. Identified variants were annotated using Ensembl VEP as well as multiple databases, including dbSNP, dbNSFP, GnomAD, ClinVar and HGMD [23–28]. Moreover, an inhouse databases of Polish WES (N>2000) was used to identify sequencing artifacts as well as variants common in Polish population.

               All variants were filtered to include those with frequency <0.01 and a predicted effect on protein sequence (unless they were already annotated as pathogenic in ClinVar or HGMD). The filtered variants were manually inspected and evaluated against patient’s phenotype and the ACMG pathogenicity criteria as implemented in Varsome [29,30]. There were no incidental findings  eligible for reporting [31]. A supplementary table with the variants pathogenicity criteria is available in the supplementary files.” 

Point 3: I do not understand what you mean in lines 38-39 by gestalt diagnosis and dysmorphology pearl s- how are these examples of targeted tests?

Response 3: We have deleted those phrases and changed the sentence to:  “Some of genetic syndromes associated with severe symptoms in neonatal period and infancy are readily diagnosed using targeted tests when symptoms are characteristic for a specific disorder”. (lines 37-38).

Point 4: Line 47: references 5 and 6 for diagnostic yield are both for WGS, not WES. There are many WES studies that could be cited.

Response 4: extra references have been cited:

  1. Clark, M.M.; Stark, Z.; Farnaes, L.; Tan, T.Y.; White, S.M.; Dimmock, D.; Kingsmore, S.F. Meta-analysis of the diagnostic and clinical utility of genome and exome sequencing and chromosomal microarray in children with suspected genetic diseases. NPJ Genomic Medicine 2018, 3, doi:10.1038/s41525-018-0053-8.
  2. Charng, W.-L.; Karaca, E.; Akdemir, Z.C.; Gambin, T.; Atik, M.M.; Gu, S.; Posey, J.E.; Jhangiani, S.N.; Muzny, D.M.; Doddapaneni, H.; et al. Exome sequencing in mostly consanguineous Arab families with neurologic disease provides a high potential molecular diagnosis rate. BMC Medical Genomics 2016, 9, doi:10.1186/s12920-016-0208-3.
  3. Stark, Z.; Tan, T.Y.; Chong, B.; Brett, G.R.; Yap, P.; Walsh, M.; Yeung, A.; Peters, H.; Mordaunt, D.; Cowie, S.; et al. A prospective evaluation of whole-exome sequencing as a first-tier molecular test in infants with suspected monogenic disorders. Genet. Med. 2016, 18, 1090–1096, doi:10.1038/gim.2016.1.
  4. Tarailo-Graovac, M.; Shyr, C.; Ross, C.J.; Horvath, G.A.; Salvarinova, R.; Ye, X.C.; Zhang, L.-H.; Bhavsar, A.P.; Lee, J.J.Y.; Drögemöller, B.I.; et al. Exome Sequencing and the Management of Neurometabolic Disorders. The New England journal of medicine 2016, 374, 2246, doi:10.1056/NEJMoa1515792.
  5. Tan, T.Y.; Dillon, O.J.; Stark, Z.; Schofield, D.; Alam, K.; Shrestha, R.; Chong, B.; Phelan, D.; Brett, G.R.; Creed, E.; et al. Diagnostic Impact and Cost-effectiveness of Whole-Exome Sequencing for Ambulant Children With Suspected Monogenic Conditions. JAMA Pediatrics 2017, 171, 855, doi:10.1001/jamapediatrics.2017.1755.
  6. Mestek-Boukhibar, L.; Clement, E.; Jones, W.D.; Drury, S.; Ocaka, L.; Gagunashvili, A.; Le Quesne Stabej, P.; Bacchelli, C.; Jani, N.; Rahman, S.; et al. Rapid Paediatric Sequencing (RaPS): comprehensive real-life workflow for rapid diagnosis of critically ill children. J Med Genet 2018, 55, 721–728, doi:10.1136/jmedgenet-2018-105396.

Point 5:  “mostly patients of ICUs in Wroclaw”- are the other patients not in the ICU? Or they are in an ICU in a different city?

Response 5: Mostly referred to the city of WrocĹ‚aw, one patient was from a different city, but in the same voivodship that WrocĹ‚aw city. Word “mostly” was deleted, it has no impact on the paper.

Point 6: Were there any incidental findings? How were these addressed?

Response 6: In new added section (WES data analysis, line 102-103) “There were no incidental findings  eligible for reporting [31].”

Green, R.C.; Berg, J.S.; Grody, W.W.; Kalia, S.S.; Korf, B.R.; Martin, C.L.; McGuire, A.L.; Nussbaum, R.L.; O’Daniel, J.M.; Ormond, K.E.; et al. ACMG recommendations for reporting of incidental findings in clinical exome and genome sequencing. Genetics in Medicine 2013, 15, 565–574, doi:10.1038/gim.2013.73.

Point 7: Line 67: you are saying you define RWES as preparation and analysis in 48-72 hours but in your abstract you say the results took 5-14 days. Is this discrepancy because of Sanger confirmation? Please clarify the time course.

Response 7: We change the sentence in to: “R-WES was defined as a process completed within 5-14 days since sample collection and  including transport to laboratory, DNA isolation,  sequencing and first analysis of WES results. WES was performed on the proband DNA using SureSelect Human All Exon v5 (16 patients) or v7 (two patients) (Agilent Technologies, Palo Alto, CA, USA) according to manufacturer’s instruction. (lines 75-78)

Point 8: Line 84: says that ten patients had severe and rapidly progressive conditions and 8 patients had progressing and severe conditions but the course of the disease was moderate. But then 14 out of 18 died. How is the course of the disease moderate if some of these 8 patients died?

Response 8: We have deleted this fragment

Point 9: Line 87: 14 out of 18 is 77.7% not 66.7%

Response 9: corrected to 77.7%.

Point 10: Line 89: were the other pregnancies normal? This may be more appropriate as an additional column in your table instead of here

Response 10: We added a sentence in Clinical Characteristic of patients: “In eight cases the pregnancy was affected by: polyhydramnios (3/8), weak fetal movements (2/8), fetal hydrops (2/8) and intracranial cysts (1/8). Prenatal period in other presented cases was normal.” (lines 115-117). We deleted the additional column in the Table 2 about prenatal symptoms because of its weak impact for the main scope of manuscript.

Point 11: Line 99: what are “first” WES results? Do you mean preliminary?

Response 11: yes. We changed the sentence: “R-WES was performed among 18 patients suspected of having a genetic disease who were in a severe condition. The preliminary WES results were available after 5-14 days.” (lines 129-130)

Point 12: Line 100: It would be more appropriate to say that a molecular diagnosis was made

Response 12:  We have corrected, “A conclusive molecular diagnosis was made in 13 out of 18 probands, corresponding to an overall diagnostic yield of 72.2%.” (line131)

Point 13: Line 110: says the following five cases have been diagnosed but only lists four diseases

Response 13: We have corrected: “In addition, the following diseases (five cases) have been diagnosed: Schaaf-Yang syndrome (OMIM:615547), hypotonia infantile with psychomotor retardation and characteristic facies 1 syndrome (IHPRF1, OMIM:611549), surfactant metabolism dysfunction pulmonary 3 (SMPD3, OMIM:610921), nemaline myopathy (OMIM:161800) and neurofascin defect (OMIM:618356).” (lines 140-144)

Point 14: Line 154: what does extend the genetic tests mean? Do you mean WGS?

Response 14: Yes, WGS, but also others genetic tests (methylation analysis, arrayCGH). We add WGS in the bracket in the end of this sentence: “In three patients, the molecular diagnosis remains unknown. For those patients, we are planning to reanalyze the WES data or extend the genetic tests (arrayCGH, WGS).”

Point 15: Line 156: are you trying to say that WES should only be done on children less than 1? What is the purpose of this sentence?

Response 15: No, the meaning was about the onset of the diseases. Sentence changed to: “A significant number of genetic diseases start in the first year of life” (line 190).

Point 16: Line 161: I disagree that a distinguishing feature of this manuscript is that the patients have severe disease requiring ICU care (PMIDs 30049826, 29644095, 31246743)- mortality rate is acceptable as a distinguishing feature

Response  16: We agree with the Reviewer. We  changed the sentence: “Our clinical sample is limited in size, but what enhances our work is the diagnostic utility and a high mortality rate [35–37].”

  1. Miller, N.A.; Farrow, E.G.; Gibson, M.; Willig, L.K.; Twist, G.; Yoo, B.; Marrs, T.; Corder, S.; Krivohlavek, L.; Walter, A.; et al. A 26-hour system of highly sensitive whole genome sequencing for emergency management of genetic diseases. Genome Med 2015, 7, doi:10.1186/s13073-015-0221-8.
  2. Farnaes, L.; Hildreth, A.; Sweeney, N.M.; Clark, M.M.; Chowdhury, S.; Nahas, S.; Cakici, J.A.; Benson, W.; Kaplan, R.H.; Kronick, R.; et al. Rapid whole-genome sequencing decreases infant morbidity and cost of hospitalization. NPJ Genom Med 2018, 3, doi:10.1038/s41525-018-0049-4.
  3. Sanford, E.F.; Clark, M.M.; Farnaes, L.; Williams, M.R.; Perry, J.C.; Ingulli, E.G.; Sweeney, N.M.; Doshi, A.; Gold, J.J.; Briggs, B.; et al. Rapid Whole Genome Sequencing Has Clinical Utility in Children in the PICU. Pediatr Crit Care Med 2019, 20, 1007–1020, doi:10.1097/PCC.0000000000002056.

Point 17: Line 166: if mortality is 14/18 that is 77.7%

Response 17: corrected; “The difference in diagnostic yield may result from a statistical fluctuation due to the relatively low number of patients. However, the main reason may be the severity of clinical condition in our cohort exemplified, among others, by the high death rate of 77.7%.” (lines 196-199)

Point 18: Line 164: how would a low number of patients change the diagnostic yield?

Response 18: We now write: “The difference in diagnostic yield may result from a statistical fluctuation due to the relatively low number of patients. However, the main reason may be the severity of clinical condition in our cohort exemplified, among others, by the high death rate of 77.7%.”

Point 19: Line 166 “notably, while in other studies patients had been more diverse…” what do you mean by diverse? In their ethnic background? Disease presentation? Then you only cite one study, not “studies”

Response 19: We changed this sentence adding: “Notably, while in Meng et al. study patients had more diverse disease in terms of severity, in the group of most seriously-ill infants the diagnostic yield had been significantly higher than in other groups [40].”

Meng, L.; Pammi, M.; Saronwala, A.; Magoulas, P.; Ghazi, A.R.; Vetrini, F.; Zhang, J.; He, W.; Dharmadhikari, A.V.; Qu, C.; et al. Use of Exome Sequencing for Infants in Intensive Care Units: Ascertainment of Severe Single-Gene Disorders and Effect on Medical Management. JAMA Pediatr 2017, 171, e173438, doi:10.1001/jamapediatrics.2017.3438.

Point 20: Line 168: further increase your diagnostic yield compared to what? The Meng study that you cite in the previous line was also WES this is confusing

Response 20: Thank you for pointing out this mistake. Tthe sentence “Furthermore, all our patients were studied by WES which is likely to further increase the diagnostic yield” has been deleted.

Point 21: Line 173: how do you say mainly caused by mitochondrial disease when it is 46.7%? This sentence doesn’t seem to add to the purpose of the paper

Response 21: in the group of 8 patients with IEM, 7 of them had a mitochondrial disease which consistent with the literature.

“In this study, IEM was found in eight patients and, consistent with the literature, it was mainly accounted for by mitochondrial disorders– (7/8) [41,42]”.

Point 22: Line 210: you have not demonstrated that your patient lacked cognitive function- they were an infant

Response 22: We changed cognitive for DEVELOPMENTAL.

“We propose that the dysfunction of its binding partner DCAF5 caused by the de novo mutation in our patient might have similar consequences for developmental function” (lines 247-248)

Point 23: Line 238: are you limiting this just to infants?

Response 23: corrected to: “Rapid-WES is an effective and time-saving diagnostic tool for infants and children in ICUs, who present heterogeneous and severe symptoms.” (lines 277-278)

Point 24:Table: what are the asterisks indicating in patients 4, 5, 8, 9, and 13

Response 24: An asterisk designates STOP codon, however to avoid misunderstanding, we have changed an asterisk into “Ter” for STOP codon

Point 25: Consider changing “lack” in column disease/omim for the undiagnosed patients to something like “non-diagnostic”

Response 25: corrected, thank you.

Point 26: These sentences in particular don’t make sense in English and need to be rewritten

  1. Line 17: “presented heterogeneous and severe symptoms”
                   Ad.1       “R-WES proved to be an effective diagnostic tool for critically ill infants in ICUs suspected for a                                 genetic disorder.”
                   2.            Line 47: “clinical examination and interview are precisely collected”- are you trying to say        that your analysis pipeline is phenotype-driven?

               Ad.2. qualifying a patient was based on the criteria presented in Table 1

  1. Line 176: fast-fatal ending

               Ad.3. “This could be in particular the case of the most severe conditions, in which a lethal ending makes the              diagnosis difficult”

  1. Line 200: vestigial spontaneous motoric

               Ad.4. “Our patient had a serious, severe and lethal disorder with the following symptoms: respiratory           insufficiency, no muscle tension and weak reflexes including sucking reflex, residual spontaneous motoric        (weak movement of upper and lower limbs), no eye contact, symmetric extensor dominance, areflexia and          absence of motor and oral automatism as well as chest deformation.”

  1. Line 218: provide a diagnostic response to their families

               Ad.5 “Finding a disease-causing mutation responsible for patients’ clinical condition is an important aspect                for the patients family.”

  1. Lines 224-225

               Ad. 6. The sentence was deleted. It had no impact for our paper.

Response 26: We corrected this mentioned phrases.

Point 27:  These sentences are also poorly written

  1. Line 6: …genetic and phenotype heterogeneity and very often rapid course…
                   Ad 1.” Genetic disorders are the leading cause of infant morbidity and mortality. Due to the large number of genetic diseases, genetic molecular and phenotype heterogeneity and often severe course, these diseases      remain undiagnosed”
                   2.            Line 20: “allows to offer”
                   Ad 2. “The quickly obtained diagnosis impacts patient’s medical management and families can receive          genetic counseling.”
                   3.            Line 32: do you mean long-term?
                   Ad 3. “Firstly, if any treatment is available, it could be implemented allowing at the same time to avoid other            ineffective or potentially harmful therapies. Secondly, long-term strategies (surgeries, rehabilitation etc.)        could be scheduled to prevent complications of the disease”
                   4.            Line 41: do you mean severely ill?
                   Ad 4. The main challenges are due to the critical illness often makes it difficult to make a diagnosis and the              early age of onset means that some patients may not have fully grown into their phenotypes to make a clear   clinical diagnosis
                   5.            Line 65: question use of the word obligatorily
                   Ad 5. The word obligatory was deleted. “Venous blood samples were collected from 18  children, from their parents and siblings (if present).”          
  2. Line 222: a full genetic consulting could be offered- do you mean consultation?

               Ad 6. “Finally, after examining parents, a full genetic consultation could be offered to the family and help the parents with further reproductive choices.”
Response 27 and 28: The mentioned phrases were changed. The text was again corrected by native English speaker.

Reviewer 3 Report

Dear Authors,

Thank you for submitting your manuscript entitled "Rapid whole-exome sequencing as a diagnostic tool in a neonatal/pediatric intensive care unit". 

Congratulations on putting this work together.  I had the pleasure to read your manuscript which would benefit greatly if you could address my points below:

  • please use italic font when referring to genes (eg. DCAF5)

Line 10-11: The abstract would benefit greatly by clarifying this was done R-WES was done on ingletons.  The sentence "Blood samples were also collected from parents" can be confusing leading the readers to believe this R-WES was done as a trio.

Line 29: please change "the total group of affected patients is large" to "the total number of affected individuals is large"

Line 37-42: in this paragraph you describe reasons why making a genetic diagnosis in critically ill infants is challenging and limits the applications of targeted sequencing.  You have not mentioned the two main challenges which are: the critical illness often makes it difficult to make a diagnosis and the early age of onset means that some patients may not have fully grown into their phenotypes to make a clear clinical diagnosis.  Please add these two factors to your paragraph. 

Line 43: Introduction paragraph starting with line 43, you rightfully describe the utility of rapid WES and WGS in the genetic diagnosis of critically ill children.  However you do not mention the turnaround time of diagnostic rate of rapid WGS.  there are good paper describing such efforts such as PMID: 30049826.

Methods section: Please indicate this was done in hospital/diagnostic lab setting or (research setting?) and how was the study funded.

Line 51: this should be "patient recruitment".  Please add a paragraph describing inclusion/exclusion criteria.  

Line 56: you use "R-WES" abbreviation without defining it in the text before hand.  Please write "Rapid-WES (R-WES)" 

Line 59: please change "genetic examination" to "genetic testing"

Line 64: "genetic analysis" please also describe how variants were filtered and whether a Software was used for that. Please describe how variant pathogenicity was assessed (by a clinical scientist? or else?).

Line 71: Rapid Run mode and analyzed as previously described [7].  Methods in reference 7 is actually a referral to methods to another reference, in which alignment was made to hg19; whereas in this paper alignment is done to hg38.  Please provide sufficient and accurate method describing WES data analysis, it is important since the core of your paper is based on WES. Also describe whether CNVs and SVs were investigated.

Table 1: please indicate what the abbreviation "F" and "DNR" refer to (eg. 2F" in patients 2 and 3, and DNR protocol).  I suggest to change: DECISION AFTER DIAGNOSIS to "IMPACT ON CLINICAL MANAGEMENT"

Line 99: please clarify how turn around time of 5-14 days was calculated: from time of patient recruitment? or from time of sequencing?

Line 124-125: you mention there is no Lo variants in DCAF5 in gnomAD. Please double check this.  There are at least 6 stop gain LoF variants reported in gnomAD V3.

Line 137: please change all NARS to NARS1 to avoid confusion with NARS2 gene.

Line 116 and 137: please change "DCAF5 gene" to "DCAF5 variant" and same for NARS.  This is because you are discussing the variant rather than the gene.  I understand you discuss the gene function and relation to patient phenotypes in the discussion section. 

Best wishes.

Round 2

Reviewer 2 Report

My concerns have all been addressed.